# A Novel *Bifidobacterium longum* Subsp. *longum* T1 Strain from Cow’s Milk: Homeostatic and Antibacterial Activity against ESBL-Producing *Escherichia coli*

**DOI:** 10.3390/antibiotics13100924

**Published:** 2024-09-27

**Authors:** Andrey V. Machulin, Vyacheslav M. Abramov, Igor V. Kosarev, Evgenia I. Deryusheva, Tatiana V. Priputnevich, Alexander N. Panin, Ashot M. Manoyan, Irina O. Chikileva, Tatiana N. Abashina, Dmitriy A. Blumenkrants, Olga E. Ivanova, Tigran T. Papazyan, Ilia N. Nikonov, Nataliya E. Suzina, Vyacheslav G. Melnikov, Valentin S. Khlebnikov, Vadim K. Sakulin, Vladimir A. Samoilenko, Alexey B. Gordeev, Gennady T. Sukhikh, Vladimir N. Uversky, Andrey V. Karlyshev

**Affiliations:** 1Federal Service for Veterinary and Phytosanitary Surveillance (Rosselkhoznadzor) Federal State Budgetary Institution “The Russian State Center for Animal Feed and Drug Standardization and Quality” (FGBU VGNKI), 123022 Moscow, Russia; 2Skryabin Institute of Biochemistry and Physiology of Microorganisms, Federal Research Center “Pushchino Scientific Center for Biological Research of Russian Academy of Science”, Russian Academy of Science, 142290 Pushchino, Russiasuzina_nataliya@rambler.ru (N.E.S.);; 3Kulakov National Medical Research Center for Obstetrics, Gynecology and Perinatology, Ministry of Health, 117997 Moscow, Russia; t_priputnevich@oparina4.ru (T.V.P.); a_gordeev@oparina4.ru (A.B.G.);; 4Institute for Biological Instrumentation, Federal Research Center “Pushchino Scientific Center for Biological Research of Russian Academy of Science”, Russian Academy of Science, 142290 Pushchino, Russia; 5Blokhin National Research Center of Oncology, Ministry of Health, 115478 Moscow, Russia; 6Alltech Company, 105062 Moscow, Russia; tpapazyan@alltech.com; 7Federal State Budgetary Educational Institution of Higher Education, St. Petersburg State University of Veterinary Medicine, 196084 Saint Petersburg, Russia; 8Gabrichevsky Research Institute for Epidemiology and Microbiology, 125212 Moscow, Russia; 9Institute of Immunological Engineering, 142380 Lyubuchany, Russia; 10Department of Molecular Medicine, Morsani College of Medicine, University of South Florida, Tampa, FL 33612, USA; vuversky@usf.edu; 11Department of Biomolecular Sciences, School of Life Sciences, Chemistry and Pharmacy, Faculty of Health, Science, Social Care and Education, Kingston University London, Kingston upon Thames KT1 2EE, UK; a.karlyshev@kingston.ac.uk

**Keywords:** *Bifidobacterium longum* subsp. *longum*, antagonistic activity, ESBL-producing *E. coli*, homeostasis

## Abstract

**Background/Objectives:** The global emergence of antibiotic-resistant zooanthroponotic *Escherichia coli* strains, producing extended-spectrum beta-lactamases (ESBL-E) and persisting in the intestines of farm animals, has now led to the development of a pandemic of extra-intestinal infectious diseases in humans. The search for innovative probiotic microorganisms that eliminate ESBL-E from the intestines of humans and animals is relevant. Previously, we received three isolates of bifidobacteria: from milk of a calved cow (BLLT1), feces of a newborn calf (BLLT2) and feces of a three-year-old child who received fresh milk from this calved cow (BLLT3). Our goal was to evaluate the genetic identity of BLLT1, BLLT2, BLLT3 isolates using genomic DNA fingerprinting (GDF), to study the tolerance, adhesion, homeostatic and antibacterial activity of BLLT1 against ESBL-E. **Methods:** We used a complex of microbiological, molecular biological, and immunological methods, including next generation sequencing (NGS). **Results:** GDF showed that DNA fragments of BLLT2 and BLLT3 isolates were identical in number and size to DNA fragments of BLLT1. These data show for the first time the possibility of natural horizontal transmission of BLLT1 through with the milk of a calved cow into the intestines of a calf and the intestines of a child. BLLT1 was resistant to gastric and intestinal stresses and exhibited high adhesive activity to calf, pig, chicken, and human enterocytes. This indicates the unique ability of BLLT1 to inhabit the intestines of animals and humans. We are the first to show that BLLT1 has antibacterial activity against ESBL-E strains that persist in humans and animals. BLLT1 produced 145 ± 8 mM of acetic acid, which reduced the pH of the nutrient medium from 6.8 to 5.2. This had an antibacterial effect on ESBL-E. The genome of BLLT1 contains ABC-type carbohydrate transporter gene clusters responsible for the synthesis of acetic acid with its antibacterial activity against ESBL-E. BLLT1 inhibited TLR4 mRNA expression induced by ESBL-E in HT-29 enterocytes, and protected the enterocyte monolayers used in this study as a bio-model of the intestinal barrier. BLLT1 increased intestinal alkaline phosphatase (IAP) as one of the main molecular factors providing intestinal homeostasis. **Conclusions:** BLLT1 shows promise for the creation of innovative functional nutritional products for humans and feed additives for farm animals that will reduce the spread of ESBL-E strains in the food chain.

## 1. Introduction

The global emergence of antibiotic-resistant zooanthroponotic *Escherichia coli* strains, producing extended-spectrum beta-lactamases (ESBL-E) and persisting in the intestines of farm animals, has now led to the development of a pandemic of extra-intestinal infectious diseases in humans. The pandemic was first announced in 2008, and it has flourished to the present day [1,2,3,4,5,6,7,8].

Pathogenic polyresistant ESBL-E strains isolated from broilers, mainly CTX-M type and belonging to the ST 131 clone, had genetic similarities and shared virulence genes with ESBL-E found in human feces. These strains were etiological agents of various extracellular infectious pathologies in humans [9]. In various countries, poultry serves as a reservoir for ESBL-E pathogens [10]. These bacteria survive at all stages of broiler production [11,12,13,14,15,16,17,18,19,20,21,22]. Other farm animals (including pigs and dairy animals), due to their direct connection to the human food chain, represent a niche and function as a depot and means for transmission and dissemination of ESBL-E pathogens [20,23,24,25,26,27]. Bovine mastitis leads to significant financial losses due milk waste and the premature culling of affected animals [28,29,30,31,32,33,34]. In recent decades, ESBL-E pathogens have settled in the human gut. They have also begun to inhabit hospital environments [35,36,37].

ESBL-E cause avian colibacteriosis, which leads to high mortality (up to 53.5%) among young chickens and costs the poultry industry hundreds of millions of dollars in economic losses worldwide [38].

ESBL enzymes produced by polyresistant *E. coli* strains hydrolyze a wide range of beta-lactam antibiotics, including oxyimino-cephalosporins and third-generation monobactams [39]. The rapid spread of ESBL-producing representatives of the *Enterobacteriaceae* family is due to the association of β-lactamase genes with conjugative plasmids [40], which in the last few decades have ensured the occupation of almost all ecological niches in the countries of five continents [4,41,42,43]. ESBL-E live in humans, animals and the environment [44,45]. Once in the intestine, ESBL-E grow intensively and produce lipopolysaccharide (LPS), which causes an increase in intestinal permeability and the entry of pathogens into mesenteric lymph nodes and the blood. This is accompanied by infection of parenchymal organs and the development of numerous extra-intestinal infectious diseases from lactation mastitis, cystitis, pyelonephritis [30,46,47,48,49], to septicemia [50]. In women, cystitis and pyelonephritis most often develop as a result of direct penetration of the pathogen from the intestine into the urogenital system [51]. Women of reproductive age turned out to be the most sensitive to the action of ESBL-E. The main niche for the habitat of ESBL-E is the intestine [52]. Penetrating from the intestine into the urinary system, ESBL-E induce the development of cystitis, pyelonephritis and stimulate premature birth in pregnant women [53,54], which has been confirmed experimentally on mice [55].

Pregnant women belong to a high-risk group, especially susceptible to the rapid progression of sepsis with significant morbidity and mortality worldwide. In addition, maternal infections can have a serious detrimental effect on newborns: almost one million deaths of newborns annually are associated with maternal infection or sepsis [56].

Urinary tract infections (UTIs) caused by ESBL-E pathogens are among the most common diseases in the world [57,58,59,60,61,62]. Risk factors for the development of UTIs are the previous use of cephalosporins and fluoroquinolones of the third- and fourth-generations, women of reproductive age, type II diabetes, and others [57]. Carbapenems are the drug of choice for the treatment of infections caused by ESBL producers, but in many countries, there is an increase in resistance to these drugs [60], which has led to an increase in mortality up to 40% [61].

Currently, according to statistical data [63,64,65,66,67], the mortality rate of patients in intensive care units for infections caused by pathogens not producing ESBL did not exceed 3% and increased to 22% in the treatment of infections caused by pathogens producing ESBL.

In recent decades, ESBL-E has started to inhabit the hospital environment [8,68], and also settled in the human intestine [36,37,69,70]. Treatment of ESBL-E infectious diseases through the use of intensive antibiotic therapy damages the normal intestinal microbiota, violates the intestinal barrier and creates conditions for the selective growth of multi-drug resistant (MDR) pathogens [71].

Numerous approaches are being developed to reduce the prevalence of MDR ESBL-E, including usage of antibiotic-free feed additives and probiotic-based supplements for pathogen competitive exclusion [72,73,74]. The antibacterial activity of 50 *Bacillus subtilis* strains was screened against ESBL-E pathogens [75]. Due to the fact that *B. subtilis* strains have a weak adhesive interaction with enterocytes, they are of little use for the elimination of ESBL-E pathogens from the intestine. The addition of a competitive exclusion probiotic to the feed of newly hatched birds inhibited colonization and reduced the level of ESBL-E in the intestines of broilers [76]. The addition of a probiotic based on “natural living intestinal microbiota obtained from chickens that do not contain specific pathogens and produced by fermentation” (Aviguard^®^, MSD Animal Health, Boxmeer, The Netherlands) to drinking water from day 0–7, prevented colonization and reduced transmission of the *E. coli* E3827 strain, which carries the ESBL blaCTX-M-1 gene [77]. Probiotic feed additive for poultry (Gro2MAX, BioNatural America, Royal Oak, MI, USA) containing *Lactobacillus acidophilus*, *Pediococcus acidilactici*, *P. pentosaceus*, *B. subtilis*, and *Saccharomyces cerevisiae* was applied orally from birth in experimental chickens. Subsequent oral administration of the ESBL-E showed a decrease in the adhesion of this pathogen to enterocytes and a loss of the pathogen’s film-forming ability [78]. The use of probiotics with the ability of competitive exclusion is promising. However, the high variability of the obtained results indicates the need for further research. According to the “One Health” concept, implemented by WHO, FAO, UNEP, and WOAH [79], the search for probiotic strains with the ability to inhabit the intestines of animals and humans and show competitive exclusion to ESBL-producing pathogens is of considerable interest. Bifidobacteria play a significant role in the formation of a healthy intestinal microbiota in humans and animals. The protective functions of bifidobacteria are performed through mechanisms of direct and indirect antagonism against opportunistic and pathogenic bacteria, preventing colonization of the intestinal biotope and preventing their penetration into the internal environment of the body [80,81,82]. Previously, we received three isolates of bifidobacteria: from the milk of a calved cow (T1), the feces of a newborn calf (T2), and the feces of a three-year-old child who received fresh milk from this calved cow (T3). According to MALDI-TOF mass spectrometry the T1, T2, T3 isolates belonged to *Bifidobacterium longum* subsp. *longum* (BLLT1, BLLT2, and BLLT3). These results were confirmed by complete genome sequencing data for BLLT1 (GenBank, Accession number: CP090414.1). We assumed that isolate BLLT2 obtained from the feces of a newborn calf and isolate BLLT3 obtained from the feces of a child who received fresh milk represent one strain (isolate BLLT1) living in the milk of a calved cow. The purpose of this work was to evaluate the genetic identity of BLLT1, BLLT2, and BLLT3 isolates using genomic DNA fingerprinting to study the tolerance, adhesion, homeostatic, and antibacterial activity of BLLT1 against ESBL-producing *Escherichia coli*.

## 2. Results

### 2.1. Morphological Characteristics of Pure Cultures of BLLT1, BLLT2, and BLLT3 Isolates of Bifidobacteria

The light microscopy of bifidobacteria pure cultures of BLLT1, BLLT2, and BLLT3 isolates are shown in Figure 1A–C. The cultures of three isolates have homologous cell morphology. The cells are arranged singly, in pairs, can combine into small aggregates, and rarely form V-shaped structures.

### 2.2. Genomic Fingerprinting of BLLT1, BLLT2, and BLLT3 Isolates

To identify the differences between the BLLT1, BLLT2 and BLLT3 isolates, we used the method of DNA genomic fingerprinting. The results are shown in Figure 2.

Genomic fingerprint analysis showed that the DNA fragments of BLLT1, BLLT2, and BLLT3 isolates are identical in number and size. Further studies were carried out using the BLLT1 isolate (strain).

### 2.3. Tolerance of BLLT1 to Gastric and Intestinal Stresses

The results of the in vitro assessment of BLLT1 tolerance to gastric and intestinal stress are presented in Table 1. According to the data obtained, BLLT1 is highly resistant to gastric stress. After 60 min of exposure to gastric juice, the degree of resistance (RD) of BLLT1 to gastric stress was 1.1 ± 0.3. BLLT1 is highly resistant to intestinal stress. After five hours of exposure to intestinal juice, the RD of BLLT1 to intestinal stress was 3.9 ± 0.4.

### 2.4. Adhesion of BLLT1 to Enterocytes

The adhesion of BLLT1 to human HT-29, porcine IPEC-J2, chicken CPCE, and bovine BPCE enterocytes is shown in Table 2. BLLT1 demonstrated efficient adhesion to human immortalized HT-29 intestinal epithelial cells (Adhesion activity (AA): 100%; Adhesion index (AI): 34 ± 3), porcine immortalized IPEC-J2 intestinal epithelial cells (AA: 100%; AI: 25 ± 3), chicken primary CPCE intestinal epithelial cells (AA: 100%; AI: 29 ± 3), and bovine primary BPCE intestinal epithelial cells (AA: 100%; AI: 36 ± 3). These findings indicate that BLLT1 exhibits high adhesive ability to enterocytes regardless of the host species (human, pig, chicken, or bovine). The strong attachment of BLLT1 to human and animal enterocytes is necessary to ensure colonization resistance of the intestine and to perform various probiotic functions in the digestive system.

### 2.5. BLLT1 Antibacterial Activity

The antibacterial activity of BLLT1 against ESBL-E is shown in Table 3. Cultivation of ESBL-E together with BLLT1 for 24 h reduced the level of living test cultures by 6 Log. ESBL-E isolates from the urine of pregnant women (IIE PW 8015, IIE PW 8034, IIE PW 8042, IIE PW 8059, IIE PW 8123, IIE PW 8147), which were an etiological factor of cystitis and pyelonephritis with subsequent premature birth, produced various ESBL types and were highly sensitive to the antibacterial action of BLLT1 (Table 3). In a mother–child pair, it was found that ESBL-E isolate (IIE WLM8194) obtained from the milk of a mother with lactation mastitis and ESBL-E isolate (IIE NB8215) obtained from feces of a premature baby (<36 weeks) receiving infected milk produce the same type of ESBL (TEM). Both isolates were highly sensitive to the bactericidal action of BLLT1 (Table 3). ESBL-E isolate (IIE Co 8316) from the milk of a dairy cow with mastitis and ESBL-E isolate (IIE Ca 8320) and isolate from feces of a calf that received milk from this cow with mastitis produced the same type of ESBL (STX-M). ESBL-E isolate (IIE Co 8316) from milk of a lactating sow and ESBL-E isolate (IIE WP8436) from feces of a weaned piglet that received milk from this lactating sow produced the same type of ESBL (SHV). This suggests that ESBL-E enters the intestines of offspring through maternal milk in farm animals.

ESBL-E (IIE LH 8508) from the feces of a laying hen and ESBL-E (IIE Egg 8517) from eggs laid by this laying hen produced the same type of ESBL (OXA), which was highly sensitive to BLLT1 (Table 3). ESBL-E (IIE LH 8525) from the feces of a laying hen and ESBL-E (IIE Egg 8532) from eggs laid by this laying hen produced the same type of ESBL (STX-M) (Table 3). ESBL-E (IIE BC 8648, IIE BC 8650, IIE BC 8675, and IIE BC 8689) from fresh broiler carcass produced various ESBL types and were highly sensitive to the antibacterial action of BLLT1 (Table 3). Thus, BLLT1 has antibacterial effects on ESBL-E pathogens, regardless of their habitat and the production of ESBL types.

### 2.6. Acetic Acid Production by BLLT1 Strain and pH Reduction of the Nutrient Medium

BLLT1 effectively produces acetic acid. The dynamics of acetic acid production is shown in Figure 3. After 12 h of BLLT1 cultivation, the acetic acid concentration in the culture medium was 52 ± 6 mM. After 24 h of BLLT1 cultivation, the acetic acid concentration in the culture medium reached up to 145 ± 8 mM. An increase in the acetic acid concentration led to a decrease in the pH of the culture medium from 6.8 to 5.2.

### 2.7. BLLT1 Whole Genome Analysis

According to GenBank records (accession number CP090414.1), the genome of BLLT1 consists of a 2,408,132 bp chromosome with 60% GC content. It does not contain any extrachromosomal elements. There are 2063 genes, including 1993 protein encoding genes and 70 pseudogenes. It has four rRNA gene clusters. These numbers are in a good match in terms of correlation with those of the genome of a reference *B. longum* subsp. *longum* strain JCM 1217 (GenBank accession number AP010888.1) with a 2,385,164 bp genome size, 60.5% GC content, and a total of 2009 genes, of which 1924 encode proteins. The genetic map (Figure 4) constructed using a Proksee server [83] and available at https://proksee.ca/ (accessed on 27 August 2024) demonstrates unusual seemingly random distribution of GC skew+ and − regions, which in other bacteria are typically positive in the leading and negative in lagging strands of a replicon [84,85]. While the Bagel5 tool [86] failed to detect any bacteriocin-related genes, a gene encoding lactococcin 972 family bacteriocin-encoding gene was found by the NCBI GenBank annotation pipeline. No prophages were found using the PHASTEST server at https://phastest.ca (accessed on 27 August 2024) [87].

Using an in vivo model of infection, alleviated production of acetic acids was previously found to be responsible for the antimicrobial activity of particular isolates of bifidobacteria [88]. In contrast with isolates lacking this activity, such isolates carried gene clusters containing ABC-type carbohydrate transporter genes. We therefore investigated the complete genome of our BLLT1 strain and confirmed the presence of a large number of such genes in its genome. Since such genes are responsible for the assimilation of carbohydrates and the production of short chain fatty acids [89], these findings are consistent with the results of our study reporting on the association of the ability of this strain to produce acetic acid with its antibacterial activity against ESBL-E pathogens.

### 2.8. BLLT1 Inhibits TLR4 mRNA Expression Induced by ESBL-E Pathogens in Enterocytes

In vitro experiments on HT-29 enterocytes have studied the ability of BLLT1 to inhibit the expression of TLR4 mRNA induced by ESBL-E pathogens.

The results of the studies are shown in Table 4. When BLLT1 was added to enterocytes, the TLR4 mRNA expression level did not change compared with the control (*p* > 0.05). Unlike the BLLT1, all ESBL-E studied in our work significantly stimulated TLR4 mRNA expression in enterocytes (*p* < 0.01). However, the preliminary introduction of BLLT1 into the culture medium of enterocytes prevented an increase in TLR4 mRNA expression induced by ESBL-E. TLR4 mRNA expression remained at the level of intact control.

### 2.9. Protection of Intestinal Barrier in HT-29 Enterocyte Monolayers by BLLT1

The ability of BLLT1 to protect the intestinal barrier (TEER) was studied in experiments in vitro on monolayers of HT-29 enterocytes. The results of the studies are shown in Table 5. The introduction of BLLT1 into the culture medium containing a monolayer of HT-29 enterocytes did not significantly affect the TEER value compared with the intact control. In all experiments, there was a slight upward trend in the TEER value. Unlike BLLT1, all ESBL-E strains significantly reduced TEER value in the enterocyte monolayers (*p* < 0.01). The preliminary introduction of BLLT1 into a culture medium containing a monolayer of HT-29 enterocytes prevented a decrease in the TEER value.

### 2.10. Inhibition of Intestinal Paracellular Permeability in HT-29 Enterocyte Monolayers by BLLT1

The ability of BLLT1 to inhibit the intestinal paracellular permeability induced by ESBL-E was studied in the in vitro experiments on the monolayers of HT-29 enterocytes. The results of the studies are shown in Table 6. BLLT1 did not significantly affect the intestinal paracellular permeability of the HT-29 enterocyte monolayer compared to the intact control. All ESBL-E strains significantly increased the paracellular permeability of the HT-29 enterocyte monolayer compared to the control (*p* < 0.01). The preliminary introduction of BLLT1 into a culture medium containing a monolayer of HT-29 enterocytes prevented an *E. coli*-induced increase in intestinal paracellular permeability.

### 2.11. Inhibition of Zonulin Secretion in the Enterocyte Monolayers HT-29 by BLLT1

The ability of BLLT1 to inhibit zonulin secretion induced by ESBL-E strains was studied in the in vitro experiments on monolayers of HT-29 enterocytes. The results of the studies are shown in Table 7. BLLT1 had no effect on zonulin secretion by monolayers of HT-29 enterocytes compared with the intact control. ESBL-E pathogens significantly increased zonulin secretion in the monolayers of HT-29 enterocytes (*p* < 0.01). The preliminary addition of BLLT1 into a culture medium containing a monolayer of HT-29 enterocytes prevented an ESBL-E-induced increase of zonulin secretion. The inhibitory effect of BLLT1 on zonulin secretion did not depend on the source of isolation of ESBL-E strains and the production by these strains of various ESBL types.

### 2.12. BLLT1 Stimulates IAP mRNA Expression in Enterocytes

The effect of BLLT1 on the expression of IAP mRNA in HT-29 enterocytes has been studied. The results are presented in Table 8. BLLT1 increases IAP mRNA expression in HT-29 enterocytes by 24 ± 2–39 ± 5% relative to the intact control (*p* < 0.05). ESBL-E strains decrease the level of IAP mRNA expression in HT-29 enterocytes to 24 ± 2–39 ± 5% relative to the intact control (*p* < 0.01). The preliminary addition of BLLT1 into a culture medium containing a monolayer of HT-29 enterocytes prevented an *E. coli*-induced decrease in IAP mRNA expression. Moreover, the increased level of IAP mRNA expression was maintained in relation to the intact control (*p* < 0.05). Stimulating effect of BLLT1 on IAP mRNA expression did not depend on the source of isolation of ESBL-E strains and the production by these strains of various ESBL types.

## 3. Discussion

Cow’s milk containing probiotic microorganisms, in particular bifidobacteria, is an ideal food for feeding calves. Bifidobacteria stimulate earlier maturation of the digestive tract, improve immunity, reduce latency to the first feed post-weaning and increase early post-weaning feed intake and growth [90]. Bacteria associated with mother’s milk control the formation of the baby’s gut microbiota [91]. *B. longum* subsp. *longum* isolates—T1 (BLLT1) was originally obtained by us from milk of a calved cow, T2 (BLLT2) was from the feces of a newborn calf, and T3 (BLLT3) was from the feces of a three-year-old child who received fresh milk from this calved cow. The calved cow was kept on a peasant farm, which practiced the use of fresh milk as a functional nutritional product for children. The method of genomic DNA fingerprinting was used to establish genomic differences or genomic similarities between the BLLT1, BLLT2, and BLLT3 isolates. Genomic DNA fingerprinting can be considered as a promising genotypic tool for the identification of a wide range of bifidobacteria at the species, subspecies and potentially up to the strain level [92,93,94]. Genomic fingerprint analysis showed that DNA fragments of BLLT1, BLLT2, and BLLT3 isolates are identical in number and size. The data obtained allowed us to assume that strain *B. longum* subsp. *longum* T1 (isolate BLLT1), which we obtained initially from the milk of a calved cow, entered with milk into the intestines of a newborn calf (BLLT2) and into the intestines of a three-year-old child who received fresh milk from this calved cow (BLLT3). In this regard, we used BLLT1 in further studies.

*B. longum* subsp. *longum* represents one of the most prevalent bifidobacterial species in the infant, adult and elderly human and animal gut [95,96,97,98], unlike *B. longum* subsp. *infantis*, which has co-evolved with humans [99]. *B. longum* subsp. *infantis* has host-specific nature. Previously, before the widespread use of antibiotics across the world, this bacterium was one of the first and most common colonizers of the intestine in infants, but not in animals [100]. This would explain why *B. longum* subsp. *infantis* grows better in the presence of human-milk oligosaccharides than in lactose [101]. *B. longum* subsp. *infantis* demonstrated its anti-inflammatory impact on fetal human enterocytes using TLR4 receptors [102]. The widespread use of antibiotics used as growth stimulants and drugs for the prevention and treatment of infections in animal husbandry, which is the basis of the human food chain containing trace amounts of antibiotics, has led to the elimination of *B. infantis* from the human intestine in the world [103]. This indicates the need to replace antibiotics with safer biological alternatives [104]. Previously, the presence of bifidobacteria in women’s milk was established [105]. Bifidobacteria in human breast milk were assessed by PCR-denaturing gradient gel electrophoresis (PCR-DGGE), and their levels were estimated by quantitative real-time PCR (qRTi-PCR). Sequences with similarities above 98% were identified as *B. breve*, *B. adolescentis*, *B. longum*, *B. bifidum*, and *B. dentium* [106].

During the breastfeeding periods for children and young animals, membrane digestion is the main one [107]. Membrane digestion is provided by bifidobacteria. The fecal microbiota of a breastfed baby contains 75–90% bifidobacteria [108]. The resistance of probiotic microorganisms to gastric and intestinal stress is an essential condition for their normal functioning in the digestive system of the host organism. BLLT1 is highly resistant to gastric and intestinal stresses (Table 1).

Adhesion to the surface of intestinal enterocytes is a necessary condition for bifidobacteria to be actively involved in the work of the digestive system and to show a beneficial effect on the host [109]. It is known that *B. longum* subsp. *infantis*, in the presence of human-milk oligosaccharides in the nutrient medium, exhibits high adhesive activity to human enterocytes, but not to those of animals [100,110]. The BLLT1 strain exhibits high adhesive activity to human, pig, chicken, and calf enterocytes (Table 2) Unlike *B. longum* subsp. *infantis*, which colonizes only the intestines of children, but not adults, *B. longum* subsp. *longum* is able to colonize both infants and adults [111,112,113]. Oki et al. conducted a longitudinal study of the duration of intestinal persistence of *B. longum* subsp. *longum* strains from infancy after birth and established the persistence of the observed strains in the intestine of a 6-year-old child [114].

Asahara et al. investigated the antimicrobial properties of probiotic bifidobacteria using the Shiga toxin-producing *E. coli* O157:H7 mice in vivo model [115]. In the infected control groups, all animals died. In the experimental group, before infection with the pathogen, a strain of *B. breve* strain Yakult was added to drinking water to colonize the intestines of mice. The subsequent infection of animals of the experimental group with *E. coli* O157:H7 did not lead to their death. *B. breve* strain Yakult produced a higher concentration of acetic acid of approximately 56 mM and lowered the pH from 7.15 to 6.0 as compared to the infected control group (acetic acid concentration, 28 mM; pH, 7.15). For the first time, we have shown that BLLT1 has antibacterial activity against ESBL-E strains isolated from humans and farm animals. The antibacterial activity of BLLT1 did not depend on the ESBL type produced by the pathogen (Table 3). BLLT1 is an effective producer of acetic acid (Figure 3). After 24 h of BLLT1 cultivation, the concentration of acetic acid in the culture medium reached up to 145 ± 8 mM. An increase in acetic acid content during BLLT1 cultivation leads to a decrease in the pH of the nutrient medium from 6.8 to 5.2. This had an antibacterial effect on ESBL-E pathogens (Table 3).

Earlier, Fukuda et al. established that some strains of bifidobacteria prevent infection with *E. coli* O157, inducing death (preventive), while other strains do not prevent infection (non-preventive). In experiments in vivo, a high concentration of acetic acid was found in the feces of mice treated with preventive strains of bifidobacteria and a low concentration of this metabolite in the feces of mice injected with non-preventive strains [88]. In bifidobacteria, short-chain fatty acids, including acetic acid, are the main end products of carbohydrate metabolism. Preventive bifidobacteria contained cassette-type carbohydrate transporter genes, responsible for the production of acetic acid inhibiting translocation of *E. coli* O157:H7 from the intestine into the blood [89]. Analysis of the complete genome of the BLLT1 strain allowed us to identify the presence of ABC-type carbohydrate transporter genes responsible for the synthesis of acetic acid, which have antibacterial activity against ESBL-E pathogens (Figure 4). Henrick B.M. and co-authors [116] found a tendency to increase the pH of feces in breastfed infants over the last century (from 5.0 to 6.5), which is associated with the loss of specialized types of bifidobacteria and represents an increased risk of intestinal dysbiosis. Acetic acid is utilized by bacteria producing butyrate in the intestine, such as *Faecalibacterium prausnitzii* [117]. Enterocytes assimilate butyrate as an energy source. Butyrate participates in mechanisms that ensure the preservation of intestinal barrier function [118] and inhibits inflammation in the intestine [119].

The discovery of Toll-like receptors (TLRs) has greatly advanced our understanding of how the body senses pathogen invasion, triggers innate immune responses, and primes antigen-specific adaptive immunity [120]. The TLR family in mammals consists of 13 members. TLRs are representatives of pattern recognition receptors (PRR). TLRs recognize evolutionarily conservative components of bacteria, viruses, fungi, and parasites, which are collectively called pathogen-associated molecular patterns (PAMP). Recognition of PAMP by PRR triggers rapid inflammatory reactions necessary for innate immunity [121,122,123,124,125]. Although TLRs are crucial for the protection of the body, evidence has accumulated that they are closely related to the pathogenesis of inflammatory diseases [126,127,128]. TLR4 is a powerful factor of innate immunity that causes inflammation in response to LPS produced by bacteria, in particular *E. coli* [129,130]. LPS is an agonist of TLR4 [130]. Diphosphorylated LPS, like *E. coli* LPS (O111:B4), is one of the most potent agonists of TLR4, whereas dephosphorylated LPS species lose their proinflammatory activity [131]. The dephosphorylation of LPS is carried out by IAP [132]. BLLT1 controls the level of IAP production by enterocytes (Table 8).

TLR4 activates the MyD88-signaling pathway at the plasma membrane [133,134,135] and after a CD14-dependent endocytosis initiates the TRIF-dependent cascade [136]. Via activated NF-κB, TLR4 also contributes to the activation of the cytosolic NLRP3 inflammasome [137,138,139,140,141].

Uncontrolled activation of TLR4 leads to excessive production of proinflammatory cytokines and the development of a cytokine storm that leads to apoptosis and the reduced proliferation of enterocytes, destroying the intestinal barrier and affects intestinal homeostasis [142,143,144]. Inflammatory signaling triggered by TLR4 during infection can lead to sepsis, septic shock, and death [129]. Infections of Gram-negative bacteria, including *E. coli* and *Pseudomonas aeruginosa*, are the predominant cause of severe sepsis in humans [145]. Low doses of LPS obtained from the gut microbiota can, under certain conditions, enter the bloodstream and cause metabolic endotoxemia, leading to low-grade chronic TLR4-dependent inflammation, which contributes to the development of metabolic diseases such as type 2 diabetes mellitus (T2DM) and obesity [146,147,148,149]. These diseases have now acquired the character of a pandemic [150]. A high level of intestinal IAP is protective against (T2DM) and obesity [151]. BLLT1 increases IAP mRNA expression in enterocytes (Table 8). The enzyme IAP is one of the main molecular factors providing intestinal homeostasis [152,153,154]. IAP dephosphorylate LPS, which loses the properties of a TLR4 agonist [132], is necessary to increase the efficiency of elimination of enteropathogens from the intestine, including ESBL-E [155]. IAP controls the development of inflammation [156] and ensures the safety of the intestinal barrier functions [157,158].

Numerous studies are currently underway to develop TLR4 antagonists [159]. Strict requirements are imposed on the TLR4 antagonists being developed. The TLR4 antagonist must maintain the intestinal barrier and ensure intestinal homeostasis [159]. BLLT1 meets these requirements. It inhibits TLR4 mRNA expression induced by ESBL-E in the HT-29 enterocytes (Table 4), protects the intestinal barrier in HT-29 enterocyte monolayers (Table 5), inhibits the intestinal paracellular permeability and zonulin secretion induced by ESBL-E in the enterocyte monolayers (Table 6 and Table 7), and increases IAP mRNA expression in enterocytes. Thus, the BLLT1 strain, obtained initially from the milk of a calved cow, has homeostatic properties and antibacterial activity against ESBL-E.

## 4. Materials and Methods

### 4.1. Bacterial Strains and Growth Conditions

The full list of bacteria used in the work and the conditions of their reproduction are given in the Table 9.

### 4.2. Assessment of the Morphology of Bifidobacteria BLLT1, BLLT2, and BLLT3 Isolates

Cell cultures were stained with methylene blue and the morphology of *Bifidobacterium* isolates was studied using light microscopy.

### 4.3. Genomic Fingerprinting of B. longum Isolates

DNA from bifidobacteria isolates was obtained by the method described in the work [160]. The strain-specific identity and/or difference of the analyzed isolates was carried out by genomic fingerprinting using ERIC primers specific to repetitive intergenic consensus sequences of enterobacteria, BOXA1 and BOXS1 primers specific to repetitive BOX sequences, as well as using REP primers specific to highly conservative extragenic palindromic repeats with a length of 35-40 base pairs [161]. The use of primers complementary to certain repetitive motifs leads to selective amplification of individual sections of the genome localized between repetitive elements. Thus, the resulting genomic fingerprints make it possible to identify differences between bacteria at the strain level.

The following primer systems proposed in the work [161] were used:

BOX A1 5′-CTACggCAAggCgACgCTgACg-3′

BoxS1 5′-CggCAAggCgACgCTgACg

ERIC1R 5′-ATgTAAgCTCCTggggATTCAC-3′

ERIC 2 5′-AAgTAAgTgACTggggTgAgCg-3′

REP 1R 5′-IIIICgICgICATCIggC-3′

REP 2I 5′-ICgICTTATCIggCCTAC-3′

### 4.4. Determination of BLLT1 Strain Tolerance to Gastric and Intestinal Stresses

Assays for gastric stress imitation and intestinal stress imitation in vitro were performed as previously described in [162].

The degree of resistance to gastric or intestinal stress RD (Resistance Degree) was determined by the formula:$$RD=\frac{n_{1}}{n_{2}},$$
where *n*_1_—the number of colony-forming cells per ml (CFU/mL) in the control and *n*_2_—(CFU/mL) in the experiment.

The level of discrimination of gastric and intestinal stress was evaluated as “very good” when the RD (discrimination ratio) is less than 5, “good” when the RD is between 5 and 10, “acceptable” when the RD is between 10 and 15, and “unacceptable” when the RD exceeds 15.

### 4.5. Enterocytes and Growth Conditions

#### 4.5.1. Human Enterocytes

A human immortalized large intestine epithelial HT-29 cell line, a relevant in vitro model system for intestinal pathogen–host cell interactions, was used. A culture medium consisting of DMEM, 10% fetal calf serum (FCS), and 0.02% penicillin and streptomycin each was used for the cultivation of the HT-29 cells. The HT-29 cells were seeded into 12-well cell culture plates at a density of 5 × 10^5^ cells/mL and were incubated for 48 h at 37 °C under 5% CO_2_ to establish a cell monolayer. To study the monolayer of HT-29 cells as an intestinal barrier, HT-29 cells were cultured at 37 °C under 5% CO_2_ for 15 days with daily replacement of the culture medium.

#### 4.5.2. Porcine Enterocytes

Porcine immortalized intestinal epithelial cells, IPEC-J2, were employed as a relevant in vitro model system for porcine intestinal pathogen–host cell interactions [163]. A culture medium consisting of DMEM, 10% FCS, and 0.02% penicillin and streptomycin each was used for the cultivation of the IPEC-J2 cells. The IPEC-J2 cells were seeded into 12-well cell culture plates at a density of 5 × 10^5^ cells/mL and were incubated for 48 h at 37 °C under 5% CO_2_ to establish a cell monolayer.

#### 4.5.3. Chicken Enterocytes

Chicken primary cecal enterocytes (CPCE) were isolated from the cecal of 2-week-old chicks (Cobb-500 cross) using a well-established protocol [164]. A culture medium consisting of DMEM, 2.5% FCS, 0.1% insulin, 0.5% transferrin, 0.007% hydrocortisone, 0.1% fibronectin, and 0.02% penicillin and streptomycin each was used for the cultivation of CPCE cells. The CPCE cells were seeded into 12-well cell culture plates at a density of 5 × 10^5^ cells/mL and were incubated for 48 h at 37 °C under 5% CO_2_ to establish a cell monolayer.

#### 4.5.4. Bovine Enterocytes

Bovine primary colon enterocytes (BPCE) were obtained from the colon of a calf using the method developed by [165]. A culture medium consisting of DMEM, 5% FCS, 1% L-glutamine, 0.1% epidermal growth factor (EGF), 0.1% each of insulin, human transferrin and 1% each penicillin and streptomycin. The BPCE cells were seeded into 12-well cell culture plates at a density of 5 × 10^5^ cells/mL and were incubated for 48 h at 37 °C under 5% CO_2_ to establish a cell monolayer.

### 4.6. BLLT1 Adhesion

Immortalized human HT-29 and porcine IPEC-J2 enterocytes, primary chicken cecal CPCE enterocytes, and bovine colon BPCE enterocytes were used as in vitro intestinal bio-models. The enterocytes were seeded into cell culture plates (2 × 10^5^ cells/mL) and incubated 48 h at 37 °C under 5% CO_2_.

The nutrient medium was removed and BLLT1 at a concentration of 10^9^ CFU/mL was added to the wells. The plates were incubated at 37 °C, under 5% CO_2_ for 2 h. The nutrient medium in the wells was replaced with a sterile PBS to remove non-adherent bacteria. Adherent bacteria was stained with azure–eosin and were quantified using a Leica DM 4500 microscope (Leica, Calgary, AB, Canada) containing the Leica IM modulator applications system (Leica, Calgary, AB, Canada). The adhesion activity and the adhesion index were determined. The adhesion activity of BLLT1 is the percentage of enterocytes on the surface of which BLLT1 cells are found. The adhesion index is the number of BLLT1 cells adhered to the surface of one enterocyte.

### 4.7. Determination of BLLT1 Antagonistic Activity

The antagonistic activity of the BLLT1 strain to ESBL-E pathogens was determined by co-cultivation in *Bifidobacterium* broth medium (HiMedia, Mumbai, India) anaerobically at 37 ± 1 °C for 24 h as described in [166]. The counting of ESBL-E cells (CFU/mL) grown in monoculture (control) and grown on LB agar in the presence of BLLT1 was carried out after 24 h by serial dilution.

### 4.8. Acetic Acid Determination in Culture Liquid of BLLT1

The suspension of BLLT1 cells grown in the broth for bifidobacteria (HiMedia, Mumbai, India) was centrifuged, the culture supernatant was filtered using a 0.22 µm pore size filter (Millipore, Burlington, MA, USA). The concentration of acetic acid in the culture supernatant was determined according to the method in [167] using HPLC with an ultraviolet–visible detector (Shidmadzu, Kyoto, Japan) set at 220 nm and a Luna C18(2) column (150 mm × 4.6 mm, 5 µm; Phenomenex, Torrance, CA, USA). HPLC-grade acetic acid was used as the standard.

### 4.9. RNA Extraction, cDNA Synthesis, and Quantitative Real-Time Polymerase Chain Reaction

Trizol reagent (Invitrogen, Carlsbad, CA, USA) was used to extract total RNA from the HT29 cells. The RNA purity was assessed spectrophotometrically by measuring absorbance at 260 nm and its ratio relative to that at 280 nm. RNase-free DNase I (Thermo Fisher Scientific, Waltham, MA, USA) was used for total RNA treatments to eliminate DNA contaminants. The integrity of RNA was monitored through electrophoresis on an agarose gel stained with GelRedTM (Biotium, Hayward, CA, USA). Subsequently, cDNA was synthesized from total RNA using the QuantiTect^®^ Reverse Transcription kit (Qiagen, Germantown, MD, USA), according to the manufacturer’s recommendations. qRT-PCR was conducted utilizing SYBR^®^ Premix Ex TaqTM (Takara Biotechnology, Otsu, Japan) on a thermal cycler (StepOnePlusTM; Applied Biosystems, Foster City, CA, USA) through 40 cycles. The reaction mixture (20 µL) comprised 5 µL cDNA, 10 µL Power SYBR^®^ Green PCR master mix (Applied Biosystems, San Francisco, CA, USA), 4 µL RNasefree water, and 0.5 µL each of forward and reverse primers. The PCR protocol involved an initial denaturation cycle at 95 °C for 10 min, followed by 40 cycles at 95 °C for 30 s and 60 °C for 30 s, with a final extension step for 30 s at 72 °C. The results were expressed as mean values averaged from three independent experiments, with the β-actin gene (ACTB) serving as an endogenous control. In all tests, a negative control was implemented to detect contamination. The relative quantity (RQ) of gene expression for the sample was calculated using the 2^−ΔΔCt^ method.

Primers used for evaluating the effects of BLLT1 on the expression of TLR4 and IAP genes in HT-29 enterocytes are shown in Appendix A. The β-actin gene was used as a housekeeping reference gene.

### 4.10. Co-Culture of HT-29 Enterocytes with BLLT1 and ESBL-E Strains to Quantify the Levels of TLR4 and IAP mRNA Expression

Enterocytes HT-29 cells were plated onto 12-well plates at a density of 10^6^ cells per well in DMEM, supplemented with 10% FCS and 0.02% penicillin and streptomycin each. Following a 24 h incubation period at 37 °C with 5% CO_2_, the culture medium was exchanged with Roswell Park Memorial Institute (RPMI) 1640 medium without penicillin/streptomycin. In experimental group 1, enterocytes were co-incubated with BLLT1 at a concentration of 1 × 10^8^ CFU/mL for 30 min at 37 °C under anaerobic conditions. In experimental group 2, enterocytes were infected with ESBL-E at a concentration of 1 × 10^7^ CFU/mL for 30 min at 37 °C under anaerobic conditions. In experimental group 3, enterocytes were co-incubated with BLLT1 at concentration of 1 × 10^8^ CFU/mL for 30 min at 37 °C under anaerobic conditions, and then were infected with ESBL-E at a concentration of 1 × 10^7^ CFU/mL for 30 min at 37 °C under anaerobic conditions. In plates with enterocytes of control group RPMI 1640 without penicillin/streptomycin was added. The cells were incubated under anaerobic conditions. After incubation, the enterocytes from the experimental groups were washed thrice with RPMI 1640 containing penicillin/streptomycin. The cell pellets of the experimental and control groups were washed twice with sterile PBS solution (pH 6.7) to quantify the levels of TLR4 and IAP mRNA expression.

### 4.11. TEER Measurements

The HT-29 cells were seeded onto apical inserts (AP) (Millicell-24, Millipore, Burlington, MA, USA) at a density of 10^5^ cells/insert (0.5 mL) and allowed to differentiate over the next 15 days. The HT-29 monolayers were pretreated with BLLT1 and then infected with ESBL-E or only infected with ESBL-E with a multiplicity of infection (MOI) of 100. After 12 h the cell monolayer integrity was characterized using an epithelial volt-ohm-meter (EVOM WPI, Friedberg (Hessen), Germany). Researches were conducted according to the manufacturer’s instructions.

### 4.12. Paracellular Permeability Evaluation

The assessment of HT-29 monolayers paracellular permeability was carried out as described earlier in our work [168].

### 4.13. Zonulin Quantification

The concentration of soluble Zonulin in the supernatants of HT-29 monolayers was evaluated using a special ELISA kit (AssayGenie, Dublin, Ireland) in accordance with the manufacturer’s instructions.

### 4.14. Statistical Analysis

The data were analyzed by one-way ANOVA and represents means ± standard deviation (SD) from six independent experiments, each tested in triplicate. Statistical significance was assessed using Student’s *t*-tests, and results were considered significant at *p* < 0.05.

## 5. Conclusions

BLLT1 is resistant to gastric and intestinal stresses and exhibits high adhesive activity to calf, pig, chicken, and human enterocytes. This indicates the ability of BLLT1 to inhabit the intestines of animals and humans. For the first time, it was shown that BLLT1 has antibacterial activity against the spectrum of ESBL-E pathogens isolated from animals and humans. The antibacterial activity of BLLT1 was associated with the effective production of acetic acid and a decrease in the pH of the nutrient medium from 6.8 to 5.2. The study of BLLT1 strain complete genome revealed the presence of ABC-type carbohydrate transporter gene clusters responsible for the synthesis of acetic acid with its antibacterial activity against ESBL-E pathogens. BLLT1 controls the ESBL-E-stimulated intestinal inflammation by inhibiting TLR4 mRNA expression in enterocytes. In enterocyte monolayers, BLLT1 protects the intestinal barrier functions, inhibits paracellular permeability, and zonulin secretion. BLLT1 increases the expression of IAP mRNA in enterocytes, which is one of the main molecular factors that ensures intestinal homeostasis. In future, we plan to conduct preclinical and clinical studies of BLLT1 according to the established medical and veterinary requirements, and create functional nutritional products for humans and feed additives for farm animals based on it in order to implement a strategy to eliminate of ESBL-E through the food chain. The effect of BLLT1 on the intestinal microbiota of broilers and the effectiveness of elimination of ESBL-E pathogens from the intestines of chickens will be studied. As a functional nutritional product for humans, a yogurt will be created containing *Streptococcus thermophilus*, fermenting cow’s milk and providing the technological parameters of a yogurt and a BLLT1 responsible for the functional properties of the innovative food product. Freeze-dried BLLT1 will be used as a feed additive for broiler chickens and other types of farm animals.

## Figures and Tables

**Figure 1 antibiotics-13-00924-f001:**
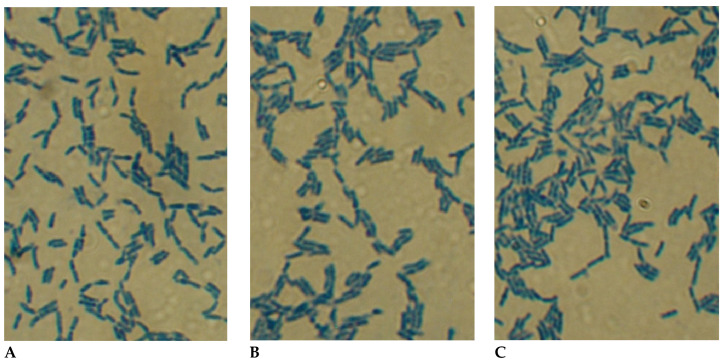
The light microscopy of pure cultures of *B. longum* subsp. *longum* isolates. (**A**)—BLLT1 obtained initially from the milk of a calved cow. (**B**)—BLLT2 obtained from the feces of a newborn calf. (**C**)—BLLT3 obtained from the feces of a three-year-old child who received fresh milk of a calved cow for 1 month. The magnification is 1350×. The cells were stained with methylene blue.

**Figure 2 antibiotics-13-00924-f002:**
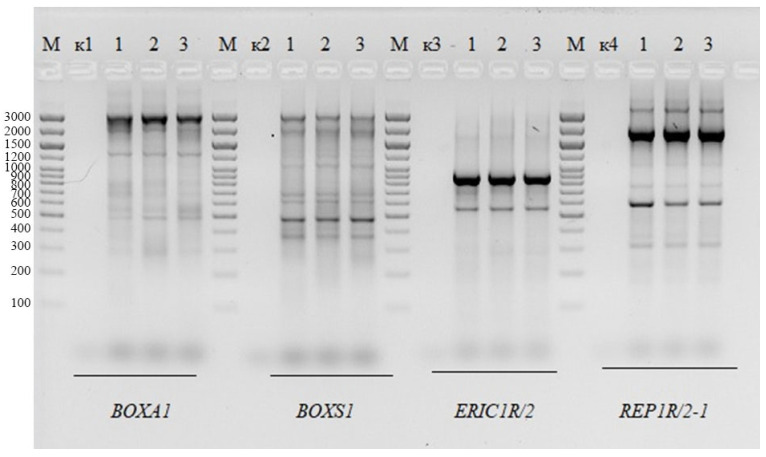
Analysis of PCR products using electrophoresis in 2% agarose gel. M—marker of molecular weight 100 bp. k—control PCR in the absence of template DNA. 1—BLLT1; 2—BLLT2; 3—BLLT3.

**Figure 3 antibiotics-13-00924-f003:**
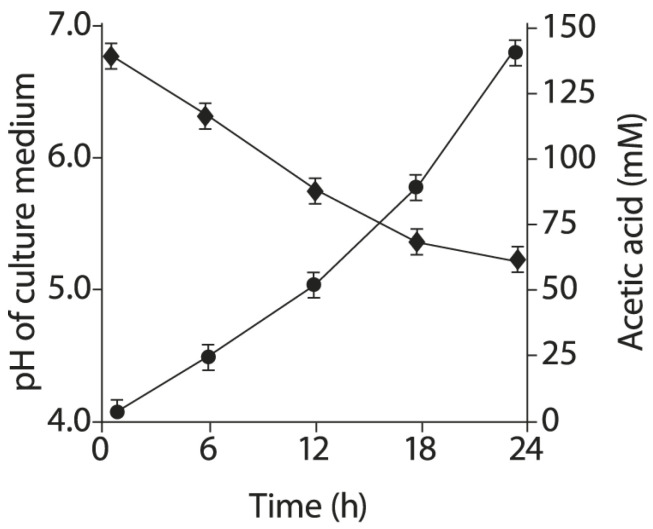
The dynamics of acetic acid production by the BLLT1 strain (●) and a decrease of the culture medium pH (♦). Data are presented as the means ± SD of six independent experiments, tested in triplicate.

**Figure 4 antibiotics-13-00924-f004:**
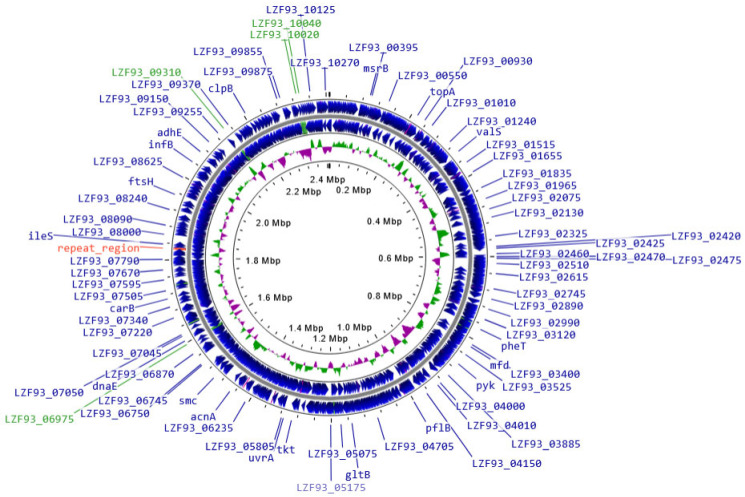
Genetic map of *B. longum* subsp. *longum* strain T1 (BLLT1) genome. GC skew+ and GC skew- strands in an inner circle are designated with green and pink, respectively. Positions of the four rRNA gene clusters are indicated by green lines. Position of a highly repetitive 5.2 kb region (at 1,824,994–1,830,209 bp) with an unknown function is indicated by a red line.

**Table 1 antibiotics-13-00924-t001:** Tolerance of BLLT1 to gastric and intestinal stresses in vitro.

Gastric Stress *						Intestinal Stress *	
10 min		30 min		60 min		5 h	
Experiment, ×10^7^ CFU/mL	Control, ×10^7^ CFU/mL	Experiment, ×10^7^ CFU/mL	Control, ×10^7^ CFU/mL	Experiment, ×10^7^ CFU/mL	Control, ×10^7^ CFU/mL	Experiment, ×10^7^ CFU/mL	Control, ×10^8^ CFU/mL
2.85 ± 0.58	3.14 ± 0.52	2.92 ± 0.49	3.21 ± 0.55	2.84 ± 0.44	3.16 ± 0.47	2.95 ± 0.62	1.15 ± 0.51
RD = 1.1 ± 0.2 Very good		RD = 1.1 ± 0.1 Very good		RD = 1.1 ± 0.3 Very good		RD = 3.9 ± 0.4 Very good	

* Data are presented as the means ± SD of six independent experiments, tested in triplicate.

**Table 2 antibiotics-13-00924-t002:** Adhesion of BLLT1 to human, porcine, chicken, and bovine enterocytes.

Enterocytes	BLLT1 Adhesion Indicators	
	Adhesion Activity (%)	Adhesion Index
Human HT-29	100	34 ± 3
Porcine IPEC-j2	100	25 ± 3
Chicken CPCE	100	29 ± 3
Bovine BPCE	100	36 ± 3

Data are presented as the means ±SD of six independent experiments, tested in triplicate.

**Table 3 antibiotics-13-00924-t003:** Antibacterial activity of BLLT1 against ESBL-E pathogens.

Strain	SLBL Type	0 h		24 h	
		C ^1^	JC ^2^	C ^1^	JC ^2^
*E. coli* ATCC BAA 2326	STX-M-15	1 × 10^6^	2 × 10^6^	9 × 10^8^	<10^2^
*E. coli* ATCC BAA 204	SHV-2	2 × 10^5^	3 × 10^5^	8 × 10^8^	<10^2^
*E. coli* ATCC BAA 198	TEM-26	3 × 10^5^	3 × 10^5^	7 × 10^8^	<10^2^
*E. coli* IIE Wo 8015	OXA	2 × 10^5^	3 × 10^5^	7 × 10^8^	<10^2^
*E. coli* IIE Wo 8034	SHV	1 × 10^6^	1 × 10^6^	6 × 10^8^	<10^2^
*E. coli* IIE Wo 8042	STX-M	4 × 10^5^	4 × 10^5^	8 × 10^8^	<10^2^
*E. coli* IIE Wo 8059	TEM	2 × 10^5^	2 × 10^5^	7 × 10^8^	<10^2^
*E. coli* IIE Wo 8123	TEM	3 × 10^5^	3 × 10^5^	8 × 10^8^	<10^2^
*E. coli* IIE Wo 8147	STX-M	2 × 10^5^	2 × 10^5^	7 × 10^8^	<10^2^
*E. coli* IIE WLM 8194	STX-M	5 × 10^5^	5 × 10^5^	8 × 10^8^	<10^2^
*E. coli* IIE NB 8215	TEM	3 × 10^5^	3 × 10^5^	8 × 10^8^	<10^2^
*E. coli* IIE NB 8224	STX-M	4 × 10^5^	4 × 10^5^	8 × 10^8^	<10^2^
*E. coli* IIE Co 8316	STX-M	2 × 10^5^	3 × 10^5^	9 × 10^8^	<10^2^
*E. coli* IIE Ca 8320	STX-M	5 × 10^5^	5 × 10^5^	8 × 10^8^	<10^2^
*E. coli* IIE Co 8356	TEM	3 × 10^5^	3 × 10^5^	8 × 10^8^	<10^2^
*E. coli* IIE Ca 8341	TEM	2 × 10^5^	2 × 10^5^	8 × 10^8^	<10^2^
*E. coli* IIE Pi 8420	SHV	4 × 10^5^	3 × 10^5^	8 × 10^8^	<10^2^
*E. coli* IIE WP 8436	SHV	1 × 10^5^	1 × 10^5^	9 × 10^8^	<10^2^
*E. coli* IIE LH 8508	OXA	2 × 10^5^	3 × 10^5^	7 × 10^8^	<10^2^
*E. coli* IIE Egg 8517	OXA	3 × 10^5^	3 × 10^5^	8 × 10^8^	<10^2^
*E. coli* IIE LH 8525	STX-M	4 × 10^5^	4 × 10^5^	8 × 10^8^	<10^2^
*E. coli* IIE Egg 8532	STX-M	1 × 10^5^	1 × 10^5^	9 × 10^8^	<10^2^
*E. coli* IIE BC 8644	TEM	6 × 10^5^	6 × 10^5^	8 × 10^8^	<10^2^
*E. coli* IIE BC 8650	STX-M	4 × 10^5^	4 × 10^5^	8 × 10^8^	<10^2^
*E. coli* IIE BC 8675	OXA	2 × 10^5^	3 × 10^5^	8 × 10^8^	<10^2^
*E. coli* IIE BC 8689	SHV	3 × 10^5^	3 × 10^5^	8 × 10^8^	<10^2^

^1^ Control; number of ESBL-E cells in monoculture (CFU/mL). ^2^ Number of ESBL-E cells in co-culture with BLLT1 (CFU/mL). Data are presented as the means ± SD of six independent experiments, tested in triplicate.

**Table 4 antibiotics-13-00924-t004:** BLLT1 inhibits TLR4 mRNA expression induced by ESBL-E in HT-29 enterocytes.

*E. coli* Strains	TLR4 mRNA (Fold Change)			
	Control	BLLT1	*E. coli*	BLLT1+ *E. coli*
*E. coli* ATCC BAA 2326	0.9 ± 0.1	0.8 ± 0.1 *	10.1 ± 0.5 **	1.0 ± 0.2 *
*E. coli* ATCC BAA 204	1.0 ± 0.2	0.7 ± 0.1 *	10.3 ± 0.3 **	1.2 ± 0.2 *
*E. coli* ATCC BAA 198	0.8 ± 0.2	0.8 ± 0.1 *	9.1 ± 0.4 **	1.0 ± 0.1 *
*E. coli* IIE Wo 8015	0.9 ± 0.3	0.8 ± 0.1 *	10.5 ± 0.6 **	1.1 ± 0.2 *
*E. coli* IIE Wo 8034	0.9 ± 0.2	0.8 ± 0.1 *	11.8 ± 0.8 **	1.0 ± 0.2 *
*E. coli* IIE Wo 8042	0.8 ± 0.2	0.8 ± 0.1 *	10.4 ± 0.7 **	1.0 ± 0.1 *
*E. coli* IIE Wo 8059	0.9 ± 0.3	0.7 ± 0.1 *	9.4 ± 0.5 **	1.2 ± 0.2 *
*E. coli* IIE Wo 8123	0.9 ± 0.2	0.7 ± 0.1 *	11.0 ± 0.6 **	1.0 ± 0.2 *
*E. coli* IIE Wo 8147	0.8 ± 0.2	0.7 ± 0.1 *	10.9 ± 0.9 **	1.0 ± 0.1 *
*E. coli* IIE WLM 8194	1.1 ± 0.3	0.9 ± 0.2 *	9.5 ± 0.5 **	1.1 ± 0.2 *
*E. coli* IIE NB 8215	0.9 ± 0.1	0.8 ± 0.1 *	11.3 ± 0.7 **	1.3 ± 0.1 *
*E. coli* IIE NB 8224	0.9 ± 0.2	0.7 ± 0.1 *	10.1 ± 0.6 **	1.0 ± 0.2 *
*E. coli* IIE Co 8316	0.8 ± 0.1	0.7 ± 0.1 *	9.0 ± 0.4 **	1.2 ± 0.1 *
*E. coli* IIE Ca 8320	1.0 ± 0.2	0.9 ± 0.2 *	11.8 ± 0.5 **	1.3 ± 0.1 *
*E. coli* IIE Co 8356	0.8 ± 0.2	0.6 ± 0.1 *	10.2 ±0.4 **	1.3 ± 0.2 *
*E. coli* IIE Ca 8341	1.1 ± 0.2	0.7 ± 0.2 *	12.6 ±0.7 **	1.4 ± 0.2 *
*E. coli* IIE Pi 8420	0.9 ± 0.2	0.8 ± 0.1 *	11.2 ±0 6 **	1.0 ± 0.1 *
*E. coli* IIE WP 8436	0.8 ± 0.1	0.7 ± 0.1 *	10.7 ± 0.8 **	1.3 ± 0.2 *
*E. coli* IIE LH 8508	0.9 ± 0.1	0.7 ± 0.1 *	8.2 ± 0.4 **	1.1 ± 0.2 *
*E. coli* IIE Egg 8517	0.9 ± 0.2	0.8 ± 0.1 *	9.0 ± 0.5**	1.0 ± 0.1 *
*E. coli* IIE LH 8525	0.8 ± 0.1	0.7 ± 0.1 *	10.4 ± 0.6 **	1.2 ± 0.2 *
*E. coli* IIE Egg 8532	0.9 ± 0.1	0.7 ± 0.1 *	11.6 ± 0.7 **	1.0 ± 0.2 *
*E. coli* IIE BC 8644	0.9 ± 0.2	0.8 ± 0.1 *	10.1 ± 0.4 **	1.3 ± 0.2 *
*E. coli* IIE BC 8650	0.8 ± 0.2	0.7 ± 0.1 *	9.3 ± 0.6 **	1.2 ± 0.1 *
*E. coli* IIE BC 8675	1.1 ± 0.1	0.9 ± 0.1 *	12.0 ±0.7 **	1.4 ± 0.1 *
*E. coli* IIE BC 8689	1.0 ± 0.1	0.8 ± 0.1 *	11.5 ± 0.6 **	1.3 ± 0.2 *

Data are presented as the means ± SD of six independent experiments, tested in triplicate. * *p* > 0.05—Control vs. BLLT1 or Control vs. BLLT1+ ESBL-E strain; ** *p* < 0.01—ESBL-E strain vs. Control.

**Table 5 antibiotics-13-00924-t005:** BLLT1 protects the barrier function in HT-29 enterocyte monolayers from destruction by ESBL-E pathogens.

*E. coli* Strains	TEER Value (Ω × cm^2^)			
	Control	BLLT1	*E. coli*	BLLT1+ *E. coli*
*E. coli* ATCC BAA 2326	254 ± 9	267 ± 10 *	106 ± 5 **	238 ± 8 *
*E. coli* ATCC BAA 204	257 ± 11	265 ± 8 *	110 ± 6 **	242 ± 10 *
*E. coli* ATCC BAA 198	249 ± 8	260 ± 11 *	104 ± 8 **	235 ± 9 *
*E. coli* IIE Wo 8015	252 ± 9	264 ± 10 *	112 ± 5 **	232 ± 9 *
*E. coli* IIE Wo 8034	245 ± 12	258 ± 11 *	105 ± 9 **	230 ± 10 *
*E. coli* IIE Wo 8042	247 ± 8	255 ± 10 *	109 ± 7 **	229 ± 8 *
*E. coli* IIE Wo 8059	250 ± 10	268 ± 12 *	111 ± 6 **	232 ± 11 *
*E. coli* IIE Wo 8123	245 ± 7	252 ± 10 *	107 ± 5 **	234 ± 9 *
*E. coli* IIE Wo 8147	253 ± 11	274 ± 10 *	112 ± 8 **	240 ± 11 *
*E. coli* IIE WLM 8194	256 ± 9	283 ± 11 *	111 ± 9 **	238 ± 10 *
*E. coli* IIE NB 8215	248 ± 12	265 ± 10 *	108 ± 6 **	226 ± 8 *
*E. coli* IIE NB 8224	250 ± 8	267 ± 10 *	109 ± 7 **	235 ± 9 *
*E. coli* IIE Co 8316	257 ± 11	284 ± 9 *	112 ± 5 **	240 ± 10 *
*E. coli* IIE Ca 8320	259 ± 10	272 ± 11 *	108 ± 9 **	239 ± 8 *
*E. coli* IIE Co 8356	252 ± 8	279 ± 10 *	106 ± 7 **	241 ± 10 *
*E. coli* IIE Ca 8341	254 ± 12	268 ± 11 *	113 ± 6 **	235 ± 9 *
*E. coli* IIE Pi 8420	247 ± 11	269 ± 10 *	104 ± 9 **	232 ± 10 *
*E. coli* IIE WP 8436	255 ± 9	284 ± 12 *	111 ± 7 **	238 ± 11 *
*E. coli* IIE LH 8508	243 ± 11	269 ± 10 *	105 ± 8 **	229 ± 12 *
*E. coli* IIE Egg 8517	252 ± 8	271 ± 9 *	104 ± 5 **	233 ± 8 *
*E. coli* IIE LH 8525	258 ± 10	275 ± 10 *	103 ± 6 **	237 ± 10 *
*E. coli* IIE Egg 8532	240 ± 7	267 ± 11 *	99 ± 5 **	234 ± 11 *
*E. coli* IIE BC 8644	248 ± 12	273 ± 10 *	105 ± 9 **	236 ± 10 *
*E. coli* IIE BC 8650	253 ± 8	285 ± 11 *	111 ± 8 **	232 ± 11 *
*E. coli* IIE BC 8675	259 ± 8	274 ± 10 *	108 ± 9 **	241 ± 10 *
*E. coli* IIE BC 8689	250 ± 10	282 ± 11 *	104 ± 6 **	235 ± 9 *

Data are presented as the means ± SD of six independent experiments, tested in triplicate. * *p* > 0.05—Control vs. BLLT1 or Control vs. BLLT1+ ESBL-E strain; ** *p* < 0.01—ESBL-E strain vs. Control.

**Table 6 antibiotics-13-00924-t006:** BLLT1 inhibits paracellular permeability induced by ESBL-E strains in the HT-29 enterocyte monolayers.

*E. coli* Strains	Paracellular Permeability (%)			
	Control	BLLT1	*E. coli*	BLLT1+ *E. coli*
*E. coli* ATCC BAA 2326	3.2 ±0.6	2.0 ± 0.5 *	46 ± 2 **	4.8 ± 0.9 *
*E. coli* ATCC BAA 204	3.6 ± 0.8	2.6 ± 0.7 *	43 ± 2 **	4.9 ± 1.2 *
*E. coli* ATCC BAA 198	3.7 ± 0.7	2.3 ± 0.4 *	47 ± 3 **	5.5 ± 0.6 *
*E. coli* IIE Wo 8015	3.3 ± 0.5	2.4 ± 0.6 *	42 ± 4 **	4.8 ± 1.1 *
*E. coli* IIE Wo 8034	3.3 ± 0.4	2.2 ± 0.5 *	48 ± 5 **	4.5 ± 1.0 *
*E. coli* IIE Wo 8042	3.9 ± 1.0	2.6 ± 0.6 *	45 ± 2 **	4.8 ± 0.9 *
*E. coli* IIE Wo 8059	3.6 ± 0.6	2.3 ± 0.7 *	47 ± 3 **	5.4 ± 0.5 *
*E. coli* IIE Wo 8123	3.4 ±0.8	2.0 ± 0.5 *	49 ± 2 **	4.9 ± 1.1 *
*E. coli* IIE Wo 8147	3.5 ± 0.7	2.7 ± 0.8 *	42 ± 3 **	3.3 ± 1.4 *
*E. coli* IIE WLM 8194	3.1 ± 0.4	2.8 ± 0.7 *	47 ± 2 **	4.5 ± 0.7 *
*E. coli* IIE NB 8215	3.8 ± 1.0	2.4 ± 0.6 *	43 ± 4 **	4.2 ± 1.2 *
*E. coli* IIE NB 8224	3.8 ± 0.9	2.6 ± 0.7 *	46 ± 2 **	4.9 ± 0.8 *
*E. coli* IIE Co 8316	3.2 ± 0.7	2.5 ± 0.8 *	44 ± 3 **	4.4 ± 1.1 *
*E. coli* IIE Ca 8320	3.5 ± 0.8	2.2 ± 0.6 *	49 ± 2 **	3.9 ± 1.4 *
*E. coli* IIE Co 8356	3.2 ± 0.5	2.9 ± 0.9 *	47 ± 3 **	4.3 ± 1.2 *
*E. coli* IIE Ca 8341	3.9 ± 0.9	2.4 ± 0.5 *	51 ± 2 **	5.8 ± 0.9 *
*E. coli* IIE Pi 8420	3.1 ± 0.4	2.8 ± 0.8 *	45 ± 3 **	4.5 ± 0.9 *
*E. coli* IIE WP 8436	3.6 ± 0.6	2.7 ± 0.9 *	53 ± 2 **	4.8 ± 0.6 *
*E. coli* IIE LH 8508	3.8 ± 1.0	2.8 ± 0.7 *	48 ± 3 **	5.2 ± 0.8 *
*E. coli* IIE Egg 8517	3.9 ± 0.9	2.7 ± 0.6 *	46 ± 2 **	4.4 ± 1.1 *
*E. coli* IIE LH 8525	3.1 ± 0.7	2.5 ± 0.5 *	49 ± 2 **	4.8 ± 1.2 *
*E. coli* IIE Egg 8532	3.7 ± 0.9	2.5 ± 0.8 *	52 ± 2 **	5.6 ± 0.7 *
*E. coli* IIE BC 8644	3.6 ± 0.6	2.7 ± 0.7 *	47 ± 3 **	4.8 ± 0.9 *
*E. coli* IIE BC 8650	3.8 ± 1.0	2.0 ± 0.4 *	45 ± 3 **	4.9 ± 1.2 *
*E. coli* IIE BC 8675	3.2 ± 0.7	2.7 ± 0.8 *	50 ± 2 **	5.3 ± 0.6 *
*E. coli* IIE BC 8689	3.0 ± 0.5	2.7 ± 0.9 *	44 ± 3 **	4.5 ± 1.1 *

Data are presented as the means ± SD of six independent experiments, tested in triplicate. * *p* > 0.05—Control vs. BLLT1 or Control vs. BLLT1+ ESBL-E strain; ** *p* < 0.01—ESBL-E strain vs. Control.

**Table 7 antibiotics-13-00924-t007:** BLLT1 inhibits zonulin secretion induced by ESBL-E-producing strains in the enterocyte monolayers HT-29.

*E. coli* Strains	Zonulin (ng/mL)			
	Control	BLLT1	*E. coli*	BLLT1+ *E. coli*
*E. coli* ATCC BAA 2326	1.7 ± 0.4	1.4 ± 0.2 *	15 ± 2 **	1.8 ± 0.3 *
*E. coli* ATCC BAA 204	1.6 ± 0.3	1.5 ± 0.2 *	15 ± 2 **	1.7 ± 0.3 *
*E. coli* ATCC BAA 198	1.8 ± 0.2	1.4 ± 0.3 *	14 ± 3 **	1.9 ± 0.4 *
*E. coli* IIE Wo 8015	1.7 ± 0.3	1.4 ± 0.2 *	16 ± 2 **	1.9 ± 0.2 *
*E. coli* IIE Wo 8034	1.6 ± 0.2	1.5 ± 0.3 *	15 ± 3 **	1.8 ± 0.3 *
*E. coli* IIE Wo 8042	1.6 ± 0.2	1.3 ± 0.2 *	14 ± 2 **	1.8 ± 0.2 *
*E. coli* IIE Wo 8059	1.6 ± 0.3	1.4 ± 0.2 *	15 ± 2 **	1.9 ± 0.3 *
*E. coli* IIE Wo 8123	1.7 ± 0.2	1.3 ± 0.2 *	14 ± 3 **	1.8 ± 0.3 *
*E. coli* IIE Wo 8147	1.6 ± 0.2	1.5 ± 0.2 *	15 ± 3 **	1.8 ± 0.2 *
*E. coli* IIE WLM 8194	1.6 ± 0.3	1.4 ± 0.2 *	14 ± 2 **	1.9 ± 0.3 *
*E. coli* IIE NB 8215	1.5 ± 0.2	1.3 ± 0.4 *	15 ± 3 **	1.7 ± 0.2 *
*E. coli* IIE NB 8224	1.7 ± 0.3	1.4 ± 0.2 *	14 ± 2 **	1.9 ± 0.3 *
*E. coli* IIE Co 8316	1.6 ± 0.4	1.5 ± 0.2 *	15 ± 3 **	1.9 ± 0.2 *
*E. coli* IIE Ca 8320	1.6 ± 0.3	1.5 ± 0.3 *	14 ± 3 **	1.8 ± 0.2 *
*E. coli* IIE Co 8356	1.6 ± 0.3	1.4 ± 0.3 *	15 ± 3 **	1.9 ± 0.2 *
*E. coli* IIE Ca 8341	1.7 ± 0.3	1.4 ± 0.2 *	15 ± 2 **	1.9 ± 0.3 *
*E. coli* IIE Pi 8420	1.7 ± 0.2	1.5 ± 0.2 *	15 ± 3 **	2.0 ± 0.3 *
*E. coli* IIE WP 8436	1.6 ± 0.3	1.4 ± 0.3 *	15 ± 2 **	1.8 ± 0.3 *
*E. coli* IIE LH 8508	1.5 ± 0.3	1.4 ± 0.2 *	14 ± 3 **	1.8 ± 0.2 *
*E. coli* IIE Egg 8517	1.6 ± 0.2	1.3 ± 0.2 *	15 ± 2 **	1.8 ± 0.2 *
*E. coli* IIE LH 8525	1.7 ± 0.2	1.5 ± 0.3 *	15 ± 2 **	1.9 ± 0.3 *
*E. coli* IIE Egg 8532	1.6 ± 0.2	1.4 ± 0.2 *	15 ± 2 **	2.0 ± 0.3 *
*E. coli* IIE BC 8644	1.7 ± 0.3	1.4 ± 0.3 *	16 ± 3 **	1.8 ± 0.3 *
*E. coli* IIE BC 8650	1.6 ± 0.3	1.3 ± 0.2 *	15 ± 2 **	1.8 ± 0.2 *
*E. coli* IIE BC 8675	1.7 ± 0.2	1.4 ± 0.3 *	16 ± 2 **	1.9 ± 0.2 *
*E. coli* IIE BC 8689	1.6 ± 0.2	1.5 ± 0.2 *	14 ± 3 **	1.9 ± 0.3 *

Data are presented as the means ± SD of six independent experiments, tested in triplicate. * *p* > 0.05—Control vs. BLLT1 or Control vs. BLLT1+ ESBL-E strain; ** *p* < 0.01—ESBL-E strain vs. Control.

**Table 8 antibiotics-13-00924-t008:** Effect of BLLT1 on IAP mRNA expression in enterocytes.

*E. coli* Strains	IAP mRNA of Control (%)			
	Control	BLLT1	*E. coli*	BLLT1+ *E. coli*
*E. coli* ATCC BAA 2326	100	126 ± 5 *	37 ± 4 **	120 ± 3 *
*E. coli* ATCC BAA 204	100	135 ± 3 *	33 ± 4 **	127 ± 4 *
*E. coli* ATCC BAA 198	100	128 ± 4 *	32 ± 5 **	121 ± 3 *
*E. coli* IIE Wo 8015	100	129 ± 5 *	35 ± 3 **	122 ± 3 *
*E. coli* IIE Wo 8034	100	134 ± 3 *	37 ± 3 **	128 ± 5 *
*E. coli* IIE Wo 8042	100	131 ± 3 *	33 ± 4 **	124 ± 3 *
*E. coli* IIE Wo 8059	100	125 ± 2 *	38 ± 4 **	119 ± 4 *
*E. coli* IIE Wo 8123	100	137 ± 3 *	39 ± 3 **	125 ± 3 *
*E. coli* IIE Wo 8147	100	130 ± 4 *	36 ± 4 **	124 ± 5 *
*E. coli* IIE WLM 8194	100	135 ± 4 *	38 ± 3 **	126 ± 4 *
*E. coli* IIE NB 8215	100	132 ± 4 *	37 ± 3 **	123 ± 3 *
*E. coli* IIE NB 8224	100	137 ± 4 *	28 ± 2 *	125 ± 4 *
*E. coli* IIE Co 8316	100	138 ± 2 *	36 ± 3 **	130 ± 5 *
*E. coli* IIE Ca 8320	100	129 ± 3 *	27 ± 2 **	121 ± 4 *
*E. coli* IIE Co 8356	100	133 ± 4 *	24 ± 2 **	125 ± 3 *
*E. coli* IIE Ca 8341	100	127 ± 4 *	39 ± 5 **	118 ± 3 *
*E. coli* IIE Pi 8420	100	130 ± 5 *	31 ± 4 **	122 ± 4 *
*E. coli* IIE WP 8436	100	126 ± 4 *	32 ± 5 **	119 ± 3 *
*E. coli* IIE LH 8508	100	134 ± 5 *	38 ± 3 **	127 ± 4 *
*E. coli* IIE Egg 8517	100	137 ± 5 *	34 ± 3 **	128 ± 5 *
*E. coli* IIE LH 8525	100	129 ± 3 *	31 ± 4 **	120 ± 3 *
*E. coli* IIE Egg 8532	100	136 ± 4 *	39 ± 5 **	127 ± 4 *
*E. coli* IIE BC 8644	100	125 ± 6 *	37 ± 4 **	118 ± 3 *
*E. coli* IIE BC 8650	100	128 ± 3 *	33 ± 5 **	119 ± 3 *
*E. coli* IIE BC 8675	100	133 ± 5 *	38 ± 2 **	126 ± 4 *
*E. coli* IIE BC 8689	100	139 ± 4 *	35 ± 3 **	130 ± 3 *

Data are presented as the means ± SD of six independent experiments, tested in triplicate. * *p* < 0.05—Control vs. BLLT1 or Control vs. BLLT1+ ESBL-E strain; ** *p* < 0.01—ESBL-E strain vs. Control.

**Table 9 antibiotics-13-00924-t009:** Bacteria used in this study and growth conditions.

Bacteria	Strain ESBL ^3^ Type	Growth Conditions
*B. longum* subsp. *longum*	IIE ^1^ T1	MRS ^a^ 37 °C anaerobically 24 h
*B. longum* subsp. *longum*	IIE T2	The same
*B. longum* subsp. *longum*	IIE T3	The same
*E. coli*	ATCC ^2^ BAA2326 STX-M-15	LB ^b^ 37 °C aerobically 18 h
*E. coli*	ATCC BAA 204 SHV-2	The same
*E. coli*	ATCC BAA 198 TEM-26	The same
*E. coli*	IIE PW ^4^ 8015 ^5^ OXA	The same
*E. coli*	IIE PW 8034 ^6^ SHV	The same
*E. coli*	IIE PW 8042 ^7^ STX-M	The same
*E. coli*	IIE PW 8059 ^8^ TEM	The same
*E. coli*	IIE PW 8123 ^9^ TEM	The same
*E. coli*	IIE PW 8147 ^10^ STX-M	The same
*E. coli*	IIE WLM ^11^ 8194 ^12^ TEM	The same
*E. coli*	IIE NB ^13^ 8215 ^14^ TEM	The same
*E. coli*	IIE NB 8224 ^15^ STX-M	The same
*E. coli*	IIE Co ^16^ 8316 ^17^ STX-M	The same
*E. coli*	IIE Ca ^18^ 8320 ^19^ STX-M	The same
*E. coli*	IIE Co 8356 ^20^ TEM	The same
*E. coli*	IIE Ca 8341 ^21^ TEM	The same
*E. coli*	IIE Pi ^22^ 8420 ^23^ SHV	The same
*E. coli*	IIE WP ^24^ 8436 ^25^ SHV	The same
*E. coli*	IIE LH ^26^ 8508 ^27^ OXA	The same
*E. coli*	IIE Egg ^28^ 8517 ^29^ OXA	The same
*E. coli*	IIE LH 8525 ^30^ STX-M	The same
*E. coli*	IIE Egg 8532 ^31^ STX-M	The same
*E. coli*	IIE BC ^32^ 8648 TEM	The same
*E. coli*	IIE BC 8650 STX-M	The same
*E. coli*	IIE BC 8675 OXA	The same
*E. coli*	IIE BC 8689 SHV	The same

^1^—Collection of Microorganisms at the Institute of Immunological Engineering (IIE), Lyubuchany, Moscow Region, Russia; ^2^—American Type Culture Collection, Manassas, VA, USA; ^3^—Extended-Spectrum Beta-Lactamase; ^4^—Pregnant Woman. Human clinical ESBL-E isolates from urine; ^5^—pregnant woman, cystitis, premature birth (<36 weeks); ^6^—pregnant woman, cystitis, premature birth (<35 weeks); ^7^—pregnant woman, pyelonephritis, premature birth (<34 weeks); ^8^—pregnant woman, pyelonephritis, premature birth (<33 weeks); ^9^—pregnant woman, pyelonephritis, premature birth (<32 weeks); ^10^—pregnant woman, pyelonephritis, premature birth (<32 weeks); ^11^—Woman lactational mastitis; ^12^—from milk of a woman with lactation mastitis; ^13^—Neonatal baby: ^14^—from feces of preterm born (<36 weeks); ^15^—from feces of preterm born (<32 weeks), necrotizing enterocolitis; ^16^—Dairy cow; ^17^—from the milk of a dairy cow with mastitis; ^18^—Calf; ^19^—from the feces of a calf that received milk from a cow with mastitis; ^20^—from the milk of a dairy cow with mastitis; ^21^—from the feces of a calf that received milk from a cow with mastitis; ^22^—Pig (lactating sow); ^23^—from the milk of a lactating sow; ^24^—Weaned piglet; ^25^—from the feces of a weaned piglet; ^26^—Laying hen; ^27^—from the feces of a laying hen; ^28^—Chicken eggs; ^29^—from a chicken eggs; ^30^—from the feces of a laying hen; ^31^—from chicken eggs; ^32^—from a fresh broiler carcass (ESBL-E strains: 8648, 8650, 8675, 8689). ^a^—Broth for bifidobacteria (HiMedia, Mumbai, India); ^b^—Luria-Bertani medium.

## Data Availability

The data are contained within the article and Appendix A.

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
