# Peer review of "A Novel Bifidobacterium longum Subsp. longum T1 Strain from Cow’s Milk: Homeostatic and Antibacterial Activity against ESBL-Producing Escherichia coli"

_antibiotics, 2024, doi:10.3390/antibiotics13100924_

Round 1

Reviewer 1 Report

Comments and Suggestions for Authors

The study titled “A Novel Bifidobacterium longum subsp. longum T1
Strain from Cow's Milk: Homeostatic and Antibacterial Activity Against
ESBL-producing Escherichia coli” was assessed as a comprehensive and
valuable study, yielding important results. I would like to congratulate
you on writing this interesting and well-designed scientific article.
The English used in writing the article is easy to read and technically
adequate.

This study examines the characteristics of a strain called
Bifidobacterium longum subsp. longum T1 obtained from cow's milk. It
focuses on the strain's ability to maintain internal stability and its
effectiveness in combating ESBL-producing Escherichia coli. The study
emphasizes the strain's survival in the gastrointestinal tract, its
ability to adhere to surfaces, and its potential applications in
functional nutrition and animal feed to mitigate the spread of ESBL-E.

Although the methods and materials used are appropriate for the purpose
of the study; expanding the genetic characterization of Bifidobacterium
longum subsp. longum T1 to include comparative genomic analyses with
other Bifidobacterium longum strains may strengthen the claims regarding
its properties.

Further context can be provided regarding the analysis of TLR4 and IAP
mRNA expression in HT-29 enterocytes.

Base pair information should be shown within the Figure 2. Please
revise the figure.

This study has the potential to significantly contribute to the fields
of microbiology, public health, and functional food development. The
study's positive implications for both human and animal health are
exciting, and further research could lead to impactful innovations in
these areas.

Minor correction:

Abstract:
Line 43: ESLB should be corrected

Keywords:
Line 62: ESLB should be corrected

Introduction:
Line 90: ESLB should be corrected

Line 91: for “LPS”; Please write the long form of the expression where
you first use the abbreviation.

Line 104: “one” should be used instead of “1”

Line 108: for 3rd and 4th; Please use "exponential expression"

Line 135: for 1st; Please use "exponential expression"

Line 150: ESLB should be corrected. Please check in the whole article.

Results:
Line 180: “five” should be used instead of “5”

Discussion:
Line 385: “Oki et al. (2018)”. Please check in the Discussion section.

Materials and methods:
Line 466: Please check “the PCR method described in [134] was used.”

Line 519: “5 %”; Please remove the space between.

37°C in Table 9 and in this section; Please standardize the use of
spaces between numbers and symbols.

Reviewer 2 Report

Comments and Suggestions for Authors

The manuscript presents novel and significant findings on the isolation and characterization of Bifidobacterium longum subsp. longum T1 (BLLT1) and its antibacterial properties against ESBL-producing Escherichia coli. The study addresses a critical issue related to antibiotic resistance, which remains a pressing global concern in both human health and agriculture. The isolation of BLLT1 from cow’s milk and the demonstration of its efficacy in inhibiting pathogenic ESBL-E strains provide valuable insights into potential probiotic-based interventions.

These are my suggestions for Authors:

The abstract should clearly highlight the key findings with more emphasis on novelty. It's crucial to state the practical implications of the research, especially how BLLT1 can be applied in real-world settings, like human nutrition or animal feed.

In the introduction, while the importance of Extended Spectrum Beta-Lactamase-producing Enterobacteriaceae (ESBL-E) is discussed, there could be more context on the global impact of antimicrobial resistance (AMR) in food production. This would better engage readers unfamiliar with ESBL-E.

Several methodologies, such as the cell culture conditions for HT-29 enterocytes and IPEC-J2 cells, could benefit from more detailed explanations to improve replicability. For instance, including more specifics on media formulations and conditions beyond "DMEM" would be useful​.

The description of qRT-PCR assays to quantify TLR4 mRNA expression is somewhat brief. Further details on primers, controls, and normalization techniques could help provide transparency​.

The statistical analysis is solid, but it would be helpful to present the raw p-values for key comparisons throughout the results section. Consider adding a table summarizing the statistical outcomes for major tests​.

Figures such as the ones showing acetic acid production should be labeled more clearly. It might also be useful to provide additional graphical representations of key data like the effects of BLLT1 on enterocyte permeability​​.

The discussion could benefit from a more thorough acknowledgment of the study's limitations. For example, the model systems used (HT-29 and IPEC-J2 cells) are simplified, and the applicability of the findings to actual intestinal environments in animals should be discussed.

The conclusion mentions creating products based on BLLT1 for eliminating ESBL-E in the food chain, but it would be valuable to provide a clearer outline of what further research is needed before commercial applications are possible​.

Comments on the Quality of English Language

Moderate English is required.

Reviewer 3 Report

Comments and Suggestions for Authors

The paper is well-written, with a suitable background that reflects the current state of research in this field. The study carried out is well organized, with adequate analytical methodology and good presentation of the results.

Specific points:

1. Line 87 - Please use italic style of the latin name

2. Line 14-148. Please specify the connection between the samples/subjects used ”from milk of a calved cow (T1), feces of a newborn calf (T2) and feces of a three-year-old child who received fresh milk from this calved cow (T3)”.  This can clarify the homogeneity of the isolated strains from genus Bifidobacterium. Probably the strain is the same. In Discussion sector Line 339-343 that was mentioned, but please clarify the strain origin in the beginning also. This will help for better understanding of the data flow.

3. Line 149-150 - The aim does not corresponding to the results described in the manuscript. “The aim of this work was to study the homeostatic and antibacterial activity of isolates against ESLB-producing E. coli (ESBL-E).” There are more data concerning the probiotic potential of the tested strain. Please revised the aim.

4. Please clarify the terminology homeostatic activity and its application in the current manuscript (including the title and the aim).

5. Please revised the Introduction. I recommended to add more information concerning the application of Bifidobacterium strains against ESLB-producing E. coli and their potential as probiotic candidates.

6.In the Introduction sector the authors mentioned in line 147 “Previously, we received three isolates of bifidobacteria”. Is this preliminary data part of previous published article? How the authors identified the strains as bifidobacteria ones? The authors have been started Results with (Line 152) Typing and Morphological Characteristics of Pure Cultures of T1, T2, T3 Isolates of Bifidobacteria, are these data the same as the mentioned in line 147. In this part the authors describe the identification by MALDI-TOF and complete genome sequencing, so is it the first data for identification of the used strain/strains. Please revised and clarify that point.

7. Line 166 Genomic Fingerprinting of B. longum Isolates

Please specify why did you decided to carry out genomic fingerprinting using ERIC and BOX primers. The literature cited is unrelated with the strain diversity of Bifidobacterium genus. Please support with data from the literature the applicability of the method used as a discriminating for Bifidobacterium strains.

8. Conclusion sector

“Genomic fingerprinting showed that DNA fragments of BLLT1, BLLT2, BLLT3 isolates are identical in number and size. In this regard, we used BLLT1 in further studies”. I recommend to the authors to revise this part of the conclusion. Probably the strain is the same, so this is not a result and it is not a reason to chose the strain BLLT1 for further studies.

Round 2

Reviewer 2 Report

Comments and Suggestions for Authors

/

Comments on the Quality of English Language

Editing is required.